# Cytokines as Biomarkers in Systemic Lupus Erythematosus: Value for Diagnosis and Drug Therapy

**DOI:** 10.3390/ijms222111327

**Published:** 2021-10-20

**Authors:** Helena Idborg, Vilija Oke

**Affiliations:** 1Division of Rheumatology, Department of Medicine, Karolinska Institutet, Karolinska University Hospital, 17176 Stockholm, Sweden; helena.idborg@ki.se; 2Center for Rheumatology, Academic Specialist Center, Stockholm Health Care Services, 11365 Stockholm, Sweden

**Keywords:** lupus, systemic lupus erythematosus, SLE, biomarkers, cytokines, interferons

## Abstract

Systemic Lupus Erythematosus (SLE) is a chronic autoimmune disease. The disease is characterized by activation and dysregulation of both the innate and the adaptive immune systems. The autoimmune response targets self-molecules including cell nuclei, double stranded DNA and other intra and extracellular structures. Multiple susceptibility genes within the immune system have been identified, as well as disturbances in different immune pathways. SLE may affect different organs and organ systems, and organ involvement is diverse among individuals. A universal understanding of pathophysiological mechanism of the disease, as well as directed therapies, are still missing. Cytokines are immunomodulating molecules produced by cells of the immune system. Interferons (IFNs) are a broad group of cytokines, primarily produced by the innate immune system. The IFN system has been observed to be dysregulated in SLE, and therefore IFNs have been extensively studied with a hope to understand the disease mechanisms and identify novel targeted therapies. In several autoimmune diseases identification and subsequent blockade of specific cytokines has led to successful therapies, for example tumor necrosis factor-alpha (TNF-α) inhibition in rheumatoid arthritis. Authors of this review have sought corresponding developments in SLE. In the current review, we cover the actual knowledge on IFNs and other studied cytokines as biomarkers and treatment targets in SLE.

## 1. Introduction

Systemic lupus erythematosus (SLE) is a prototype of systemic autoimmune disease with a hallmark of autoantibody production against cell nuclei (antinuclear antibodies -ANA) and other intranuclear proteins including double stranded DNA (dsDNA). The disease may affect different organs and organ systems including skin, joints, kidney, the central and peripheral nervous systems. Hematological abnormalities, such as cytopenias and coagulation disturbances with associated risk for thromboembolic events are also common.

Activation of major components of the innate and adaptive immune systems are observed in patients with SLE. Increased levels of a range of cytokines are observed during the disease course [1]. Secreted cytokines can be observed in the circulation, saliva, urine and also identified in the target organ tissues such as skin, kidney and synovia. The majority of those cytokines have pro-inflammatory properties, but some have immunomodulatory or anti-inflammatory roles. Since the pathogenesis of the disease is not well understood, it is so far not clear if high levels of certain cytokines are driving the disease, or just an epiphenomenon to the malfunctioning immune regulation and responses, cell death or impaired clearance of unviable cell remnants.

A range of cytokines have been observed to correlate with SLE disease activity and proposed as therapeutic candidates for active SLE [1,2,3]. Cytokine blockade targeting TNF-α and IL-6, have been successfully developed to treat rheumatoid arthritis, while barely any anti-cytokine therapies have been successfully introduced into clinical practice as drugs for SLE patients. The discussed pitfall in SLE clinical trials has been the recruitment of heterogenous patient populations. It has therefore been suggested to stratify patients prior to recruitment to trials in regard to clinical and genetic phenotypes or cytokine profiles [4,5].

A biomarker that aids the physician in decision making regarding diagnose and treatment would be a handy tool. Some investigators have suggested a definition of an ideal biomarker for SLE [6]. This biomarker should be specific, easy to detect, follow the disease activity, useful in follow-up and reasonably priced.

In the current review we will focus on recent advances and on evidence for the use of cytokines as biomarkers of SLE. If known, we will further discuss the role of cytokines in certain organ involvement. If available, we will include information on the outcomes of cytokine targeting therapies in clinical trials. The reviewed information on available knowledge of cytokine expression, association with disease activity and target organ involvement is presented in Table 1. The cells that are the main cytokine sources are illustrated in Figure 1. In Table 2 we have summarized the data on recent clinical trials targeting cytokines for treatment of SLE. 

## 2. Interferons

Interferons (IFNs) are pro-inflammatory cytokines produced in response to infections, as a part of the host defense by the innate immune system. In the human body, three types of IFNs are known to be produced and all seem to have role in SLE: type I, II and III. The type I IFNs include 17 subtypes of IFN-αs, also IFN-β, IFN-δ, IFN-ε, IFN-κ, IFN-τ,I FN-ω and IFN-ζ [2].

The IFN-αs are the major effector cytokines of this type and it is the most studied in SLE. Knowledge on IFN-β and IFN-κ is also available, but other type I IFNs are less studied and considered of less importance [3,9].

All IFN-αs and single IFN-β are ubiquitously expressed, while IFN-κ is more organ specific [10]. The most potent type I IFNs producing cells are plasmacytoid dendritic cells [1,4]. All type I IFNs signal via same receptor denoted IFNAR1/IFNAR2 [2].

There is only one type II IFN—namely, IFN-γ—which is mainly produced by CD4+ or CD8+ Th1 lymphocytes, as well as NK and B cells, and also professional antigen presenting cells. The IFN-γ signal is transmitted via receptor composed of two subunits IFNGR1 and IFNGR2 [9].

Four molecules, which belong to the type III IFN family, have been identified: IFN-λ1, -λ2, -λ3 and -λ4. The main producers of IFN-λ1 are antigen presenting cells, but also epithelial cells [10]. Epithelial and mucosal surfaces are the main responders and expressers of IFNλ-1R/IL10R [3].

### 2.1. Type I IFNs in SLE

#### 2.1.1. IFN-α

It was described already in 1990 that administration of IFN-α as a therapy for other diseases, can induce SLE [11]. Later on, upregulation of type I IFN signature has been discovered in SLE patients [12]. Ever since, type I IFNs have been a major focus of investigation as a diagnostic biomarker and therapeutic target for SLE [10,13].

Detection and reliable measurements of IFN-α have been challenging due to its diversity, bio-degradability ex vivo and lack of specific detecting antibodies. Several methods have been used to study IFN-α in SLE: reporter cell assays, dissociation-enhanced lanthanide fluorescence immunoassay (DELFIA) [14], single-molecule-array (SIMOA) and enzyme-linked immunosorbent assay (ELISA) [15,16,17,18]. Recently, conventional ELISA techniques have been substantially improved and utilized in several studies [5,18].

The classical method to measure serum induced type I IFN activity is by reporter cell assays (WISH cell line or healthy donor peripheral blood mononuclear cells (PBMCs)) [19]. These assays estimate how cells exposed to SLE patients’ serum in vitro respond by upregulating IFN regulated genes. The most commonly assessed genes are MX Dynamin Like GTPase 1 (*MX1*), Protein kinase R (*PKR*) and Interferon Induced Protein With Tetratricopeptide Repeats 1 (*IFIT1*).

High serum induced type I IFN activity, as observed by WISH-reporter cell assay (IFN-activity), is defined as a heritable risk factor to develop SLE later in life [19]. High IFN-activity associate with SLE disease activity as observed in European and North American cohorts [5,16,20]. A certain auto-antibody profile has been coupled with high IFN-activity, including anti-dsDNA, anti-RNP, anti-Sm and anti-Ro autoAbs, rather than the antiphospholipid antibody profile (aPL) [20,21,22]. In our Karolinska SLE cohort high functional IFN-activity correlate positively with disease activity scores (both SLEDAI and SLAM) and, also with certain organ active involvement: e.g., nephritis, arthritis, lymphadenopathy, fatigue and weight loss [5]. The IFN-activity high group commonly have autoantibodies against dsDNA, nucleosome, Sm, SmRNP and RNP68; as well as demonstrate higher ESR and higher proteinuria, but lower hemoglobin, and WBC as well as PLT counts [5]. Photosensitivity and high disease damage associate negatively. High IFN-activity seems to be a feature of active younger patients early in the disease, since this parameter is associated with younger age, shorter disease duration and negatively correlated with disease damage index(SDI) in several studies [5].

Importantly, IFN-activity correlate with serum/plasma measurements of IFN-α, IFN-γ and IFN-λ. Therefore, we consider that IFN-activity could be an important, but less specific marker of activation of several IFN types in an active patient [4,5].

Dissociation-enhanced lanthanide fluorescence immunoassay (DELFIA) is another assay employed to measure the most the IFN-α subtypes (except IFN-α2b) [4]. This method was used to measure IFN-αs in another Swedish cohort in 2000. The results demonstrated that all patients with rash, more than half of patients with lupus nephritis (LN) and two thirds with arthritis had upregulated serum levels of IFN-α [16]. IFN-α measurements were compared to serum induced IFN-activity in healthy donors (HD) PBMCs (SLE-HD-IIF-activity). SLE-IIF-activity did not correlate with serum IFN-α measurements, and only white blood cell counts, and thrombocyte counts correlated negatively with both measurements [16]. Anti-dsDNA levels and decreasing complement levels correlated with IFN-α levels (Delphia), but not with SLE-HD-IIF [14,16]. In later studies, SLE-HD-IIF method has not been a method of choice.

A French group performed digital single-molecule-array (SIMOA) enzyme-linked immunoorbent assay (ELISA) to study IFN-αs in SLE [17]. In this cohort, high IFN-α levels associated with active disease which manifested as rash, low complement, and anti-Sm autoantibodies, and numerically (but non-significantly) increased in active nephritis. Interestingly, data demonstrated that the subgroup with active arthritis had significantly lower levels of IFN-α [17].

We employed a pan-IFN-α ELISA, which detects all IFN-αs (but not IFN-α2a and IFN-α12) to study IFN-α expression in Karolinska SLE cohort [5]. We found that in the cohort high IFN-α concentrations associated with mucocutaneous involvement, lymphadenopathy, low complement, positivity for Ro52/SSA, La/SSB, and lower occurrence of secondary antiphospholipid antibody syndrome and vascular events [5]. Patients with active LN had increased IFN-α levels which declined after therapy [23].

The available techniques for type I IFN detection are still relatively laborious and costly, and non-sufficiently reproducible. A study comparing IFN-activity, DELPHIA, SIMOA and ELISA methods in a substantial number of patients is needed in order to compare and validate all the methods.

In summary, levels of circulating IFN-α have been demonstrated to correlate with disease activity as estimated by SLEDAI, and anti-dsDNA levels and complement activation or consumption. Estimation of circulating IFN-α could potentially be used as a disease biomarker. So far it does not seem to have much additive value in comparison to traditionally assessed parameters such as SLEDAI, anti-ds-DNA or complement measurements, which are relatively cheap and available methods in most parts of the world. Therefore, these methods are mainly used in research. None of the type I IFN measuring methods have so far been introduced in routine clinical care, neither for diagnostic nor for prognostic purposes.

#### 2.1.2. IFN-β

The majority of information on the role of IFN-β in SLE comes from gene expression studies, where signatures of IFN-β and IFN-αs partially overlap [15,22]. Data indicate that levels of IFN-β and IFN-α overlap [21]. Detailed knowledge on which SLE manifestations or phenotypes, if any, are associated with high circulating IFN-β levels remain to be studied. However, IFN-β signals through the IFN-α receptor (IFNAR), therefore anti-IFNAR therapies could most possibly block effects mediated by IFN-β [22].

#### 2.1.3. IFN-κ

IFN-κ is another member of the type I IFN-family, which is more limited to expression in certain tissues, e.g., in epithelial cells [24,25]. IFN-κ has been demonstrated to be upregulated in Cutaneous lupus erythematosus (CLE) lesions after skin exposure to UV-radiation [26]. Thus, IFN-κ detection could be used by pathologists in lesion diagnostics, but otherwise it is probably less suitable as disease biomarker for routine use since a skin biopsy is needed in order to assess its expression. IFN-κ could potentially serve as a treatment target for CLE, especially if a topical anti-IFN-κ treatment can be developed.

### 2.2. IFN Type II in SLE

It was described over thirty years ago that exogenously infused IFN-γ had the ability to induce SLE or exacerbate SLE flares [24,27]. Focus on IFN-γ in SLE has been limited, but several recent reports have demonstrated the importance of this mediator in SLE [20,28]. Importantly, levels of IFN-γ increase in parallel to autoantibody development, years before activation of type I IFNs and clinically manifest SLE [25]. Interest in IFN-γ increased recently due to two reasons: unsuccessful clinical trials on type I IFN-blockade and also demonstration that signature of type I, II and III IFNs overlap [25]. We and others have identified that levels of IFN-γ correlate with IFN-activity, as measured by cell reporter assays in vitro. In the Karolinska SLE cohort, high IFN-γ was associated with high SLEDAI scores, active arthritis, complement consumption and positivity for anti-Ro60/SSA [20,28]. Interestingly, mycobacterium tuberculosis IFN-γ release assay (TB-IGRA) could demonstrate spontaneous IFN-γ release (SIR) in the SLE cohort. Normalized TB-IGRA values correlated with disease activity better than anti-dsDNA or complement levels, and were associated with cutaneous disease, hypocomplementemia, fever and thrombocytopenia [29]. SIR estimated by TB-IGRA test has been suggested as SLE biomarker by the investigators [29].

In summary, recent data indicate that IFN-γ levels incline before developing SLE symptoms and high circulating levels are associated with more severe SLE. SIR, assessed by TB-IGRA test, should be further explored as a disease biomarker. Direct or indirect medications targeting IFN-γ pathway are of further interest in drug development.

### 2.3. IFN Type III in SLE

The role of type III IFNs, including four subtypes of IFN-λ1, -2, -3 and -4, has been identified during the recent years. IFN-λs are easier to study since they may be detected in circulation by conventional ELISAs or by immunohistochemistry. Levels of IFN-λ3 have been reported to correlate with SLE disease activity, active LN and arthritis, and complement consumption [28,30,31]. IFN-λ1 is upregulated locally at the site of inflammation in CLE skin and LN renal lesions [23,30,32]. Levels of IFN-α and IFN-λ1 were measured in the Karolinska LN cohort before and after induction therapy for LN [23]. Overall levels of IFN-αs decreased after therapy, but levels of IFN-λ1 decreased only in patients who responded to therapy, while they remained high in histological non-responders. Data from this study indicate that high IFN-λ1 levels in patients who did not respond to therapy could be a biomarker of therapy resistant LN. However, this data should be replicated and validated.

In our cohort, we also noted that high IFN-λ1 levels are a feature of patients with cardiovascular events, secondary antiphospholipid syndrome (APS) and often associated warfarin treatment. High IFN-λ1 with co-upregulation of Th17 cytokines identified patients with renal damage [18].

## 3. IFNs as Treatment Targets in SLE

### 3.1. IFNs Type I Targeting Therapy 

Blockade of type I IFNs has been suggested to be a promising treatment approach for SLE. Treatment modalities included either blockade of circulating IFNs or blocking of the IFN-α receptor.

Several pharmaceutical companies developed anti-type I IFN monoclonal antibodies that have been tested in phase 2 and 3 clinical trials [22,33] (Table 2). The selected patient groups were active SLE without life-threatening disease manifestations on stable “standard of care” therapies and fixed steroid doses. The typical patient recruited for the studies had a SLEDAI score of 6 or higher, was positive for anti-dsDNA antibodies and had low complement levels. The clinical manifestations were dominated by cutaneous, articular or hematological symptoms, or serological findings. However, none of the studies reached primary endpoints [22,33].

Anifrolumab, a type I interferon receptor antagonist, was designed to block signaling of all type I IFNs [22]. In clinical trials, anifrolumab treatment was demonstrated to be beneficial in moderate to severe treatment resistant non-renal SLE, and furthermore it neutralized the IFN-signature in those patients who were high at recruitment [34]. Anifrolumab has recently got approved by United States Food and Drug Administration (FDA) for treatment of moderately severe SLE and is further studied both in the trials and real life [22,35].

### 3.2. IFN-γ Targeting Therapy

Anti-IFN-γ therapy was tested in SLE patients with or without nephritis and also with a discoid form of cutaneous lupus erythematosus [7,36]. Unfortunately, effects of the therapy were less pronounced and not sustained in the subjects. Also, a majority of the patients reported adverse events [7,36]. Direct targeting of IFN-γ does not seem to be a suitable treatment approach. Levels of IFN-γ can be modulated via other mechanisms. Interestingly, in the Phase II clinical trial of ustekinumab in SLE, the responders had declining IFN-γ levels, and this finding was suggested as a biomarker of response to anti (IL-12)/IL-23 p40 therapy [37].

### 3.3. IFN-λ Targeting Therapy

No studies directly targeting IFN-λ were registered at www.clinicaltrial.gov (accessed on 10 May 2021).

## 4. TNF-α

Tumor necrosis factor α (TNF-α) is a major cytokine secreted by macrophages upon the cell encountering a pathogen. Abnormally high TNF-α production is observed in several autoimmune diseases including SLE. TNF-α can either promote or modulate autoimmunity [38]. TNF-α has been shown to be upregulated in SLE patients compared to healthy controls and circulating levels of TNF-α have been shown to correlate with disease activity [39,40,41]. High levels of circulating TNF-α associate with active renal, articular, and neuropsychiatric manifestations such as mood and anxiety disorders, but was not significantly different with regard to active CLE [41,42]. However, TNF-α is upregulated in CLE lesions and its expression levels in the tissue correlate with the development and healing of UV-induced lesions [42,43].

In our Karolinska SLE cohort we found that ratio of TNF-α and serum albumin levels were the best discriminators between patients and controls, in particular those with renal involvement [40]. There is now a consensus that high levels of circulating TNF-α are associated with SLE and might be a useful biomarker [13].

Attempts to block TNF-α in SLE were not as successful as in rheumatoid arthritis and resulted in exacerbations of the disease as reviewed by De Bandt [44]. Therefore, the rationale and evidence for TNF-blockade in SLE were reviewed as a double-edged sword [8]. Some authors consider that TNF blockade might still be beneficial in a subset of patients. According to the case reports, anti-TNF therapy seems to have a positive effect in targeting lupus arthritis, and it seems not to induce nephritis, but rather autoantibodies and rash [8]. Interestingly, data show that the mode of action of the commonly used antimalarial drugs is partly via suppressing both TNF-α and IFN-α levels [45,46]. Today, the use of TNF-blockade in SLE is still limited and not recommended as treatment due to risks.

## 5. BAFF/APRIL

B cell activating factor (BAFF), also known as B lymphocyte stimulator (BLyS), and A proliferation-inducing ligand (APRIL) are members of the TNF superfamily and the role of the BAFF/APRIL system in SLE has been reviewed in detail before [47,48]. BAFF are increased in patients with SLE and have been reported to predict flares [49]. Several, but not all studies demonstrated correlation between serum levels of BAFF/APRIL and disease activity, as measured by the British Isles Lupus Assessment Group (BILAG) index [50] and SLEDAI [47,48]. Possibly high levels of BAFF/APRIL are more strongly linked to specific organ involvements, such as arthritis, neurologic manifestations or renal disease [48]. Studies of these biomarkers in urine are limited, although elevated levels of BAFF in urine have been detected in a fraction of patients with lupus nephritis [51].

Belimumab, a BAFF inhibitor, was approved for SLE indication in 2011. It is the only biological drug approved for treatment of SLE patients so far [52]. Belimumab is a monoclonal antibody that inhibits BAFF and it has been demonstrated to reduce disease activity and prevent organ damage in SLE [53,54]. Novel treatment approaches of SLE have recently been reviewed, and while Belimumab is approved, Atacicept, a combined BAFF/APRIL inhibitor [55], is being further studied in phase II/III clinical trials [56].

Serum levels of both BAFF and APRIL have been suggested as biomarkers for treatment response to atacicept [57]. However, levels of circulating BAFF and APRIL could be affected differently by immunosuppressive drugs, e.g., APRIL can decrease while BAFF can be unchanged during treatment [58]. This discrepancy should be further explored.

## 6. IL-2

Interleukin-2 is a pleiotropic cytokine that plays multiple functions in T cell homeostasis and differentiation. It is mainly produced by activated CD4+ and CD8+ cells. IL-2 has a role in T cell pathogen recognition and discrimination between “foreign and self” [59].

Impaired production of IL-2 has been observed in SLE, and IL-2 deficiency was associated with renal impairment [60]. Interestingly, SLE patients may produce anti-IL-2 autoantibodies, which have been associated with disease activity [61].

Low dose treatment with IL-2 has been tested as a therapy for active SLE in a limited number of patients. The trial did not meet the primary endpoint, i.e., did not reduce disease activity; however, significantly more nephritis patients reached remission in the IL-2, as compared to the placebo arm [62]. Further studies are needed to investigate potential benefit of IL-2 reconstitution in SLE.

## 7. IL-6

Interleukin-6 is a pro-inflammatory cytokine upregulated in several systemic autoimmune diseases including SLE [63]. Some studies demonstrated that plasma levels of IL-6 correlate to SLE disease activity, but other investigators could not confirm these findings. Interleukin-6 reflects the cytokine-mediated inflammation, which is not always followed by activation of complement. The latter is an important item in several measures of SLE disease activity, including SLEDAI [64]. Upregulated plasma levels of IL-6 have been observed in patients with kidney damage [39,40]. In a recent review, IL-6 was suggested as an important mediator in lupus nephritis [65], and urinary IL-6 has also been suggested as a marker for it [53,54].

Tocilizumab is a monoclonal antibody that inhibits IL-6 signaling and its efficacy in SLE patients was first evaluated in 2010 [66]. Tocilizumab therapy may be efficient in subgroups of patients with high inflammatory activity, although cautions must be taken since high doses may result in immunosuppression [67]. So far, no IL-6 targeting therapies have been approved for treatment of SLE.

## 8. IL-10

IL-10 is an anti-inflammatory cytokine, but it also has pro-inflammatory properties that might be mediated by type I IFNs [68]. Its role in the pathogenesis of SLE is complex. Variations in IL-10 gene are associated with risk for SLE [13,69]. Plasma levels of IL-10 are increased in SLE patients compared to controls [40], and some studies show correlation to disease activity or to levels of anti-dsDNA antibodies [51,60,61].

A trial with anti-IL-10 monoclonal antibody demonstrated reduced disease activity in SLE patients, although patients developed anti-drug antibodies and long-term treatment must be further studied [70]. A phase IIa clinical trial of an anti-IL-10 monoclonal antibody (BT063) the primary endpoint was achieved according to the company 2020 [71].

## 9. IL-16

Interleukin-16 is a cytokine, produced by T cells, epithelial cells and Langerhans cells. IL-16 is produced as a precursor molecule, which must be cleaved by caspase-3 to become activated. Interestingly, this mechanism, despite caspase-3 activation, does not necessarily induce cell apoptosis [72]. The generated product is a mature secretory IL-16 and has CD4 and CD9 as ligands. The secreted IL-16 mediates CD4+ T cell chemotaxis and functions as T cell growth factor, while cleaved pro-IL-16 mediates cell functions within the cell nuclei and inhibits cell cycle progression [73]. Interleukin-16 has been demonstrated as a promoting mediator of several cancers [73].

High levels of IL-16 in active SLE patients were described over 20 years ago [74]. Recently, our group reported results from a Mesoscale Discovery (MSD) multiplex analysis of circulating cytokines in SLE patients [40]. Among 30 tested cytokines, IL-16 was identified among the top five and increased concentrations in the circulation were found in patients with active nephritis. Also, in another unbiased analysis of cutaneous proteome of CLE lesions, IL-16 was identified as the only cytokine differentially upregulated in CLE in comparison to dermatomyositis (DM) [75]. These findings encourage further studies on the IL-16 role in SLE.

## 10. IL-12, IL-17 and IL-23

Interleukin-12 is a bridging cytokine between innate and adaptive immune systems, composed of a dimer entitled p40 and p70 and is secreted mainly by dendritic cells upon their stimulation by microbial antigens. Importantly, the p40 subunit is shared with IL-23. Interleukin-12 in parallel to IFN-γ divert T cell development towards Th1 response. SLE patients have increased levels circulating of IL-12p40 subunit, which correlate positively to disease activity and associate negatively with complement C3 levels [6,71].

Interleukin-23 is the cytokine produced by antigen presenting cells which diverts CD4+ cell development towards Th17 phenotype. Interleukin-23 is comprised of p19 and p40, the latter shared with IL-12. The IL-17 cytokine family consists of six related proteins IL-17A, B, C, D, E and F. IL-17 is mainly produced by Th17 cells, but also by, γδ-T cells, some natural killer T (NKT) cells and TCRβ+ or so called ‘natural’ Th17 cells [76]. The major biological role of IL-17 is protection against fungal and bacterial infections, but it also has a protective role in intestinal barrier integrity. Interestingly, besides IL-17, Th-17 cells also produce IFN- λ1, and together these molecules provide protection from pathogens on the skin surface [77]. Several studies reported that the majority of SLE patients have increased levels of IL-17A and IL-17F [78,79].

Levels of circulating IL-17A have been found to correlate with the Cutaneous Lupus Erythematosus activity and damage score index (CLASI) score [79], and high IL-17A expression has been demonstrated in the cellular infiltrates of the skin tissue of CLE lesions [80], as well as in kidneys affected by lupus nephritis (LN) [48]. High IL-17A levels at baseline predicted poor response to LN therapy [78] and deteriorating kidney function, if paralleled by INF-λ1 and IL-23 [18]. Also, persistently high levels of IL-23 were detected in non-responding lupus nephritis patients [78]. Increased levels of IL-17A in parallel to high IL-6 expression were found in synovial fluid derived from patients with active lupus arthritis, the same cells, when stimulated, could produce high amounts of IFN-γ [81].

Several investigators suggested that targeting IL-12, IL-23, and IL-17 could be a therapeutic option in SLE, since modulation of this pathway may also have regulatory effect on the IFN system via indirect mechanisms [82]. This hypothesis is under investigation. Ustekinumab, antagonist of p40, shared by IL-12 and IL-23, was studied in a phase 2 trial as an add-on therapy to standard of care. The study has demonstrated positive effects on clinical and laboratory parameters of SLE—namely, cutaneous and articular manifestations—with a good safety profile. The responders had declining IFN-γ levels [83]. The results of long-term outcomes are awaited.

## 11. IL-1, IL-18 and IL-38

The interleukin-1 cytokine superfamily includes several molecules: IL-1α, IL-1β, IL-18, IL-33 and IL-38. The IL-1 system is a part of the innate immune system and operates via inflammasome as a primary response of pathogen recognition. Several investigators found that peripheral levels of IL-1α were in the same range as in controls. In the Karolinska SLE cohort, however, high IL-1α levels were observed in patients with renal and articular manifestations [40]. Increased levels of tissue IL-1β were found in the skin of photoprovoked developing CLE lesions. This was accompanied by increased TNF-α and HMGB1 expression [42].

Interleukin-18 is a cytokine secreted mainly by macrophages and has an effect on IFN-γ production by Th1 cells and splenocytes, and may act in synergy with IL-12, another IFN-γ-inducible cytokine [84]. High serum levels of IL-18 are found in SLE patients, and particularly patients with active renal disease who were prone to develop renal damage at the follow-up [85]. High IL-18 expression is also observed in CLE lesions [86]. A recent metanalysis proposed IL-18 as a biomarker for active SLE [87]. In mouse models, blockade of IL-18 delays onset of the SLE-like autoimmunity and this approach is worth further exploration in humans [84].

IL-38 has been relatively recently studied in SLE. The levels of this cytokine were higher in active patients and associated with multiple clinical and laboratory parameters including arthritis, pericarditis, haematuria, proteinuria, pyuria and anti-dsDNA. Correlation analysis indicated that plasma IL-38 correlated positively with SLEDAI and negatively with C3 and C4 levels [88]. 

## 12. IL-21

Interleukin-21 (IL-21) is an autocrine cytokine mainly produced by follicular helper T (Tfh) and T helper 17 (Th17) cells. It has a role in the development of Tfh and Th17 cells, but also differentiation of B cells into plasma cells mediated by several biological pathways, including JAK/STAT [89]. Single-nucleotide polymorphisms in IL-21 and its receptor (IL-21R) have been associated with susceptibility to SLE. Increased amounts of IL-21 positive cells were reported in SLE patient circulation [90]. Other investigators found reduced levels of circulating IL-21 in SLE patients [91]. Data indicate that locally secreted IL-21 could facilitate development of autoreactive B-cells [92]. Molecules targeting IL-21 are currently tested in clinical trials for treatment of SLE (Table 2).

## 13. Conclusions

To conclude, available data demonstrate that SLE is a complex autoimmune disease and details of the underlying disease mechanisms remain to be understood. Multiple cytokines from different cytokine networks are elevated in SLE. However, data indicate that different manifestations or SLE phenotypes are associated with different cytokine networks and patterns. Thus, no single perfect disease biomarker has been identified so far. Benefits of the available cytokine targeting therapies are still limited. Available evidence indicates that different SLE phenotypes are associated with diverse cytokine networks and patient subgroups will require tailored therapies. More detailed disease phenotyping would be helpful when planning therapeutic trials and tailoring personalized treatment.

## Figures and Tables

**Figure 1 ijms-22-11327-f001:**
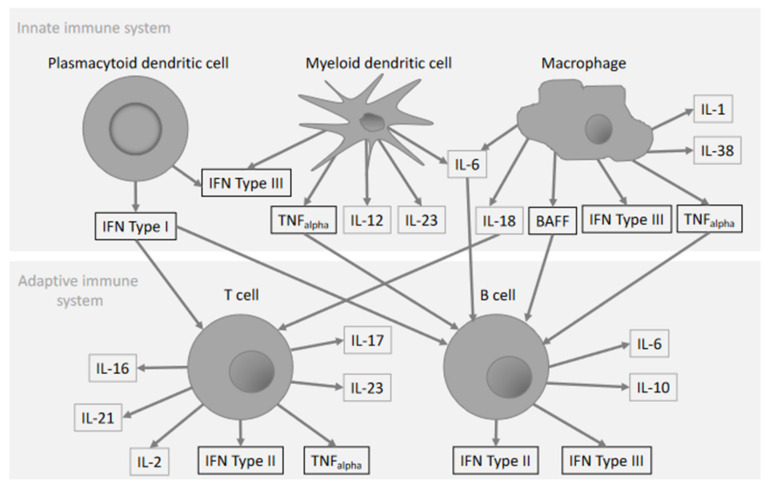
Cytokines associated with SLE pathogenesis are shown and arrows indicates the main producer or the main target cell.

**Table 1 ijms-22-11327-t001:** Overview of the cytokine regulation in the circulation, their correlation to disease activity, as well as association with and expression in the target organs of active SLE.

Cytokine	Immunity ^1^	Levels in Circulation ^2^	DAS ^3^	SLE Target Organ Involvement ^4^					Where Cytokine Has Been Detected
				LN	Arthritis	CLE	Serositis	CNS	
IFN Type I	I	↑	+	↑	↑	↑	↑	↑	circulation
IFN Type II	A	↑	+	↑	↑	↑	NN	NN	circulation
IFN Type III	I	↑	+	↑	↑	↑	NN	NN	circulation, skin lesions
BAFF/APRIL	I	↑	+	↑	↑	NN	NN	↑	circulation
IL-2	A	NN	NN	↑	NN	NN	NN	NN	renal tissue
IL-6	I	↑	+	↑	↑	↑	NN	↑	circulation, CSF, urine, serum
IL-10	I	↑	+	↑	NN	↑	NN	NN	circulation
IL-16	A/I	↑	+	↑	NN	↑	NN	NN	circulation, urine, skin lesions
IL-12, IL-23	A/I	↑	+	↑	NN	↑	NN	NN	circulation, kidney
IL-17	A	↑	+	↑	↑	↑	NN	NN	circulation, kidney, skin, synovial fluid
IL-1	I	↑	NN	↑	↑	↑	NN	NN	circulation, skin
IL-18	I	↑	+	↑	NN	↑	NN	NN	circulation, skin lesions
IL-38	I	↑	+	↑	↑	NN	↑	NN	circulation

^1^—Describes the cytokine’s involvement in the immune system: I—the cytokine is a mediator of the innate immune system; A—the cytokine is a mediator of the adaptive immune system; ^2^ ↑—upregulated cytokine levels in circulation; ^3^ DAS—disease activity score, +—cytokine is shown to be associated with DAS (SLAM or SLEDAI (SLEDAI-2K)); ^4^ If known, we mention what active SLE manifestations have been demonstrated to be associated with each cytokine; IL—interleukin, LN—lupus nephritis, CLE—cutaneous lupus erythematosus, CNS—central nervous system, CSF—cerebrospinal fluid, BAFF—B cell activating factor, APRIL—a B cell proliferation-inducing ligand, NN—not known or data is uncertain.

**Table 2 ijms-22-11327-t002:** Overview over registered therapies and ongoing trials on substances targeting cytokines as treatment targets for SLE (published or registered at www.clinicaltrials.gov, accessed on 10 October 2021. NCT number indicate registration number of clinical trial).

Cytokine Target	Drug/Molecule and Results from Clinical Trials
IFN Type I	Anifrolumab (MEDI 546), phase 3, primary endpoints met NCT01753193, approved by FDA in 2021; and studied post-registration NCT04877691. Lupuzor, IFNa kinoid (IFN-K), completed phase 2B NCT01058343, secondary endpoints were met, ongoing phase 3, NCT02665364 Sifalimumab MEDI-545, completed phase 2b, primary putcome not met, NCT00979654. JNJ-55920839, Anti-IFN-α/ω, phase 1 ongoing, NCT02609789. Rontalizumab, primary endpoint was not met, phase 2 NCT00962832
IFN Type II	AMG 811 (anti-IFN-gamma), primary endoint not met, phase 2, NCT02291588, NCT00818948 [7]. Emapalumab-Igsz (Gamifant), phase 3, has not started recruitment
IFN Type III	No trials identified
General IFN system: Target plasmacytoid dendritic cells	daxdilimab VIB7734, phase 1, completed, results awaited VIB7734 BIIB059, primary endpoints met in phase 2 NCT02847598). Phase 3 ongoing, NCT04895241
TNF-α	Infliximab, considered risky, investigated in open label NCT00368264 [8].
Blys/BAFF/APRIL	Benlysta, approved, postregistration studies ongoing Ianalumab/OP0302 (VAY736) ongoing phase 2, NCT03656562, results awaited. Rozibafusp alfa (AMG 570), results awaited, phase 2, NCT04058028 Tabalumab (LY 2127399), primary endpoints not met in phase 3, NCT01196091 Atacicept, primary endpoints not met in phase 3, NCT00624338 Blisibimod (AMG 623/A-623), primary endpoints not met, phase 3, NCT02443506 Telitacicept, RC18, Phase 2, NCT02885610
IL-2	Recombinant Human Interleukin-2, positive results, Phase 2{He:2020cn} NCT02465580 and NCT02932137 Several substances: NKTR-358 (LY3471851), ILT-101 and more, Phase 1 and 2, NCT03556007, NCT04433585
IL-6	MRA 003 US Ongoing phase 1 NCT00046774 Ala-Cpn10 Ongoing phase 1 and 2 NCT01838694 Vobarilizumab (ALX-0061) Completed phase 2, NCT02437890 Sirukumab (CNTO 136) Completed phase 1, NCT01702740, not further investigated PF 04236921, completed phase 2, primary endpoint not met, NCT01405196
IL-10	BT 063, Phase 2 completed, results unavailable, NCT02554019
IL-16	Not identified
IL-17	Sekucinumab, Phase 3 recruiting, NCT04181762
IL-12, IL-23	Ustekinumab, primary endpoints met in phase 2a, NCT02349061. Phase 3 is ongoing, NCT03517722
IL-21	NNC0114-0006, Phase 1, NCT01689025 BOS161721 (avizakimab). Completed phase 1 and 2, results awaited NCT03371251
IL-1	Anakinra, only case reports
IL-18	Not identified
IL-38	Not identified

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
