# Peer review of "Cytokines as Biomarkers in Systemic Lupus Erythematosus: Value for Diagnosis and Drug Therapy"

_ijms, 2021, doi:10.3390/ijms222111327_

Round 1

Reviewer 1 Report

The review entitled " Cytokines as Biomarkers in Systemic Lupus Erythematosus: Value for Diagnosis and Drug Therapy" presents a well-organized body of data supporting the plausibility of using cytokines as biomarkers and treatment targets in Systemic Lupus Erythematosus.  The authors sufficiently focused on recent advances on multiple cytokines in Systemic Lupus Erythematosus and their possible role as biomarkers of SLE and targeting therapies.

Minor comments:

page 3-line 104:  MX1, PKR, and IFIT1, the genes are with no explanations

page 3-line 123:  DELFIA assay mentioned with no explanations

page 5-line 208:  APS – no abbreviation

page 6-line 21:  BILAG index with no explanations

Author Response

Dear reviewer,

thank you for reading our manuscript and providing comments to improve the manuscript.

We have no addressed all the points you raised:

1. page 3-line 104:  MX1, PKR, and IFIT1, the genes are with no explanations

The explanations have been added.

2. page 3-line 123:  DELFIA assay mentioned with no explanations

The explanations have been added.

3. page 5-line 208:  APS – no abbreviation

The explanation of the abbreviation has been added.

4. page 6-line 21:  BILAG index with no explanations

The explanation has been added.

We hope you will find improved manuscript acceptable to be published in the journal.

Reviewer 2 Report

This review describes the association with cytokines (biomarker) in SLE and the value for diagnosis and drug therapy. This work is interesting to readers, however, some points should be more clearly described to understand the presented issue. 

  1. The authors show an overview of cytokine regulation, correlation with disease activity, and expression in target organs in SLE. However, the distinct regulation is a bit hard to understand only by Table1. Authors should present the association with cytokines and SLE pathogenesis (mainly presented cytokines) in a new Figure.
  2.  The authors should make another table to demonstrate treatment targets (cytokines) and expected/tried agents more clearly. If the agents are ongoing clinical trial, their information also should be included the table. 

Author Response

Dear reviewer,

Thank you for reading manuscript and providing constructive suggestions for improvement of the manuscript.

  1. The authors show an overview of cytokine regulation, correlation with disease activity, and expression in target organs in SLE. However, the distinct regulation is a bit hard to understand only by Table1. Authors should present the association with cytokines and SLE pathogenesis (mainly presented cytokines) in a new Figure.

Author response: Thank you for this recommendation. We have now added a schematic figure with information what cells are the major source and major target of the cytokines we discuss in the paper. Please see attached figure 1.

2. The authors should make another table to demonstrate treatment targets (cytokines) and expected/tried agents more clearly. If the agents are ongoing clinical trial, their information also should be included the table. 

Author response: Thank you for raising this question and this recommendation. We have now included table 2 with information covering ongoing and recently completed clinical trials on substances targeting cytokines as treatment targets in SLE. Please see the table included at the end of the manuscript.

I hope you will find improved manuscript acceptable for publication at the IJMS journal.

Sincere regards,

Vilija Oke

Reviewer 3 Report

  This interesting review article has comprehensively reviewed the current knowledge on interferons and other studied cytokines as disease biomarkers and treatment targets in systemic lupus erythematosus (SLE). The authors indicate that various manifestations or phenotypes are associated with different cytokine networks and patterns, and there are still limited benefits of the available cytokine targeting therapies in SLE. The manuscript is well written in English and the content is relevant to clinical application. There is one minor comment as follows.

  In the Introduction section lines 36-37, the authors described “Increased levels of a range of cytokines are observed during the disease course [1,2].” Nevertheless, reference 2 was the results from the OBSErve Canada Study in belimumab use in SLE patients. The author should delete this reference or replace it with another one discussing the elevated cytokine networks in SLE.

Author Response

Dear reviewer,

thank you for reading our manuscript and observing errors. We have now removed the irrelevant reference and added the correct ones.

We hope that you will find the improved manuscript now acceptable for publication.

Sincere regards,

Vilija Oke